# Acceptance and Adherence to COVID-19 Vaccination—The Role of Cognitive and Emotional Representations

**DOI:** 10.3390/ijerph19159268

**Published:** 2022-07-28

**Authors:** Simão Pinho, Mariana Cruz, Cláudia Camila Dias, José M. Castro-Lopes, Rute Sampaio

**Affiliations:** 1Department of Biomedicine, Faculty of Medicine, University of Porto, 4200-319 Porto, Portugal; m.belo.cruz@gmail.com (M.C.); jclopes@med.up.pt (J.M.C.-L.); rutesampaio@med.up.pt (R.S.); 2São João University Hospital Center, 4200-319 Porto, Portugal; 3CINTESIS@RISE, MEDCIDS, Faculty of Medicine, University of Porto, 4200-319 Porto, Portugal; camila@med.up.pt; 4Knowledge Management Unit and Department of Community Medicine, Information and Health Decision Sciences, Faculty of Medicine, University of Porto, 4200-319 Porto, Portugal

**Keywords:** COVID-19, vaccination, adherence, cognition, emotion, representations

## Abstract

Vaccine hesitation is a topic of utmost importance, with the COVID-19 pandemic serving as a clear reminder of its timeliness. Besides evaluating COVID-19 vaccine acceptance in a sample of Portuguese people, this study aims at understanding cognitive and emotional representations related to vaccination, and their influence on vaccination hesitation. A cross-sectional online survey was conducted between 27 December 2020 and 27 January 2021. It assessed cognitive and emotional COVID-19 representations; vaccination status; cognitive and emotional representations of vaccination and perceived necessity and concerns about vaccines. Of 31 × 58 participants, 91% accepted taking a COVID-19 vaccine. Among several other significant findings, women (71.3%) more often considered that the pandemic affected their lives (*p* < 0.001) and were more often concerned with being infected (*p* < 0.001). Likewise, there were significantly more female participants concerned about taking a COVID-19 vaccine and its possible effects, when compared to the number of male participants (*p* < 0.001). The number of participants with a higher education level that were more worried about becoming infected was greater (*p* = 0.001), when compared with those less educated. Regarding age groups, people aged 18 to 24 had fewer concerned participants (9.6%), while the number of individuals aged 55 to 64 had the most (*p* < 0.001). Somewhat surprisingly, perceiving oneself as extremely informed about COVID-19 was not associated with greater vaccine acceptance (OR = 1.534 [1.160–2.029]; (*p* = 0.003)). Moreover, people aged 25 to 64 years old and with lower education level were more likely not to accept vaccination (OR = 2.799 [1.085–7.221]; (*p* = 0.033)). Finally, being more concerned about taking a vaccine lowers its acceptance (OR = 4.001 [2.518–6.356]; (*p* < 0.001)). Cognitive and emotional representations have a great impact and are reliable predictors of vaccine acceptance. Thus, it is of extreme importance that public health messages be adapted to the different characteristics of the population.

## 1. Introduction

On 7 January 2020, the Severe Acute Respiratory Syndrome Coronavirus 2 (SARS-CoV-2) was first isolated and identified as the causative agent of an outbreak of pneumonia, reported in December 2019, in Wuhan, China [1]. About two months later, on 11 March 2020, the World Health Organization (WHO) declared that Coronavirus disease 2019 (COVID-19), caused by SARS-CoV-2, had reached pandemic status [2]. Since then, this pandemic has led to dramatic societal changes [3] and has had a profound economic impact [4], with no end in sight. More importantly, as of 15 November 2021, there have been 253,163,330 confirmed COVID-19 cases, causing 5,098,174 deaths, in a clear display of this disease’s great infectiousness and severity [5]. Notably, there is still no completely effective, universally accepted treatment [6]. As such, the development of effective vaccines and the successful implementation of vaccination strategies is of the utmost importance. 

Besides the challenges in minimizing COVID-19 spread and finding treatments for it, this pandemic poses new, complex problems regarding science communication to the public. Indeed, nowadays, information is easily accessible, but often manipulated, poorly interpreted or simply poorly transmitted, a phenomenon that transcends COVID-19 but is particularly impactful in this context [7]. This high prevalence of misinformation has undoubtedly contributed to the number of people that do not trust the efficacy and safety of the COVID-19 vaccines [8]. 

Vaccination distrust, which may lead to outright vaccination refusal, is not a new concept. In fact, in 2015, the WHO defined *vaccine hesitancy* as a *delay in acceptance or refusal of vaccination despite availability of vaccination services. Vaccine hesitancy is complex and context specific, varying across time, place and vaccines. It is influenced by factors such as complacency, convenience and confidence* [9,10]. The recognition of this entity’s complexity is paramount in the effort to tackle it. As such, it is fundamental to interpret vaccine hesitancy through a framework of disease and treatment representations. In this context, the Necessity–Concerns Framework [11,12], used to describe treatment engagement and adherence, is particularly useful. Accordingly, people engage in an implicit cost–benefit analysis in which beliefs about necessity are weighed against concerns about the potential adverse effects in taking it [13]. 

In Portugal, there is a consensus that vaccine acceptance is widespread. In fact, Portugal remains the country with the highest vaccine confidence in the latest *State of Vaccine Confidence in the EU + UK (2020)* report. Auspiciously, Portugal led the world in the COVID-19 vaccination rate in September of 2021 and was, as of November 2021, in third place in the world with a total of 88% of population fully vaccinated [14] 

Bearing all this in mind, the objective of the present study was to determine what was, in the beginning of the vaccination effort, the level of COVID-19 vaccine acceptance in a sample of Portuguese participants and to understand COVID-19 and vaccine representations at the time, based on the Self-Regulatory Model [15]. This evidence-based Health Psychology model postulates that, when an individual finds their health to be threatened, they find coping mechanisms by comparing their current situation to past experiences and deriving behavior patterns from them. As such, besides personal illness experience, other factors such as cultural knowledge and context cues all influence judgments made about symptom interpretation and subsequent coping behaviors. Moreover, this study intended to adapt and apply the Necessity–Concerns Framework [11] in this context, exploring the impact of representations on vaccination hesitation. The Necessity–Concerns Framework is a Psychology framework that helps describe adherence, by determining that it is a complex entity influenced by implicit judgements of an individual’s need for a certain treatment (Necessity Beliefs), which are outweighed by the same individual’s Concerns that arise from having to undergo that specific treatment, be it due to adverse effects or various personal and sociocultural consequences. In this study, the Beliefs about Medicines Questionnaire, which takes into account the Necessity–Concerns Framework and has been validated in the Portuguese population [16], was used. The choice of questionnaire was done after careful deliberation, so that we could reliably test our hypothesis that COVID-19 vaccination acceptance and adherence can be predicted by interpreting patients’ Cognitive and Emotional Representations through the lens of the Self-Regulatory Model and the Necessity–Concerns Framework. 

## 2. Materials and Methods

### 2.1. Data Collection

We conducted an anonymous online survey between the 27 December 2020 and the 27 January 2021 using the online platform *LimeSurvey*, which is an excellent, viable tool for cost-effectively collecting large amounts of data, in a relatively short period of time. Data collection began on the first day of the vaccination process in Portugal. The survey was divulged by the Communication Service of the Faculty of Medicine of Porto, as well as through 15 different online newspapers throughout the country. 

All participants were informed about the study objectives and data collection procedures. They were invited to sign a consent form, where they gave authorization for their data to be used. Likewise, the email of the research team was provided, so that any questions about the study could be answered. 

Ethical approval for the study was provided by the Hospital of S. João, Porto review boards and ethics committee (FMUP/HSJ 497/2020).

### 2.2. Inclusion and Exclusion Criteria

Inclusion criteria were that participants were aged 18 years or older at the time of the survey, resided in Portugal and were able to complete the survey in Portuguese. There were no exclusion criteria.

### 2.3. Questionnaire 

The questionnaire was constructed following CHERRIES [17] (Appendix A—CHERRIES checklist) and based on the Self-Regulation Model [15], the Necessity–Concerns Framework [11], and was divided in four parts. The first part evaluated COVID-19 representations, namely: consequences, control, concern, understanding and emotional response. The internal consistency of this scale as measured by Cronbach’s alpha is acceptable (α = 0.77). The second part assessed COVID-19 vaccination status, sorting those who had been inoculated with the vaccine and those who were not vaccinated. Unvaccinated subjects advanced to the third part of the questionnaire, regarding their intention to accept or not COVID-19 vaccination. The fourth part questioned individuals about their cognitive and emotional representations of vaccination: vaccine efficacy, worry, information and emotional response, their perceived vaccine necessity and their perceived concerns about COVID-19 vaccination. This also revealed an acceptable Cronbach’s alpha (α = 0.77). The full questionnaire is available as Appendix A—Questionnaire). 

### 2.4. Statistical Analysis 

Categorical variables were described using absolute (*n*) and relative (%) frequencies, while continuous variables were described as mean and standard deviation, or median, interquartile (IQR) range, and minimum and maximum, as appropriate. Hypotheses regarding categorical variables were tested using a Chi-square test or a Fisher’s exact test, as appropriate. The assessment of internal consistency of the questionnaires created was performed by assessing the Cronbach’s alpha statistic. 

Logistic regression was applied to determine the relationship between vaccine acceptance and cognitive and emotional representations of COVID-19, perceived vaccine effectiveness, vaccine knowledge and its acceptance. Variables such as sex, age, level of education and area of residence were used to adjust the developed models. Odds ratios (OR) and 95% confidence intervals (95% CI) were calculated. Models were built according to the Enter approach and the goodness-of-fit was assessed using the Hosmer–Lemeshow statistic. All reported *p*-values were two-sided, and the significance level was set at 5%.

Statistical analysis was performed using the software Statistical Package for the Social Sciences (SPSS) v. 26.0 (IBM Corp: Armonk, NY, USA). 

## 3. Results

### 3.1. Sociodemographic Characteristics of the Sample 

The survey was opened by 3158 individuals. However, 30 declined participating in the study and 38 did not complete the survey, bringing the sample down to a total of 3090 subjects. Their sociodemographic characteristics are presented in Table 1. Most participants were female (71%), and the mean age was 42 years old (sd = 15.22), with a minimum of 18 and a maximum of 81 years old. Participants were mainly employees (54%) and had a higher education degree. They lived predominantly in the Porto region (61%). Most were never infected with COVID-19 (94%), 5% had had the disease and 1% were infected during the assessment moment. 

### 3.2. Sociodemographic Characteristics and COVID-19 and Vaccine Representations 

Most subjects revealed a high level of perceived life disturbance due to the COVID-19 pandemic. In fact, 18% said it affected their life extremely, 48% declared it had a lot of impact, 30% answered it had a moderate impact and only 3% and 1% mentioned it had little or no impact, respectively. 

Comparing sex differences (Appendix A), significantly more female participants perceived their life to be affected (97%) by the COVID-19 pandemic, when compared to male participants (94%) (*p* < 0.001). Moreover, there were more female participants worried about possibly becoming infected (94%), when compared to men (87%) (*p* < 0.001), and a greater number of women felt affected emotionally (91%) because of the pandemic (*p* < 0.001), when compared to male participants (80%). There were likewise significantly more female participants concerned about taking a COVID-19 vaccine (48%) and about its possible effects (63%), when compared to the number of male participants, (38% and 53%, respectively) (*p* < 0.001).

Regarding education level (Appendix A), there were significant differences in the number of participants that were concerned about becoming infected in each group. The number of participants with a higher education level that were generally more worried about becoming infected was significantly greater (53%) (*p* = 0.001). There were also significant differences regarding the emotional effect of the pandemic: more individuals with a secondary education were affected (53%) (*p* = 0.039) and more felt they were more informed (58%) (*p* < 0.001). A significantly greater number of participants with a basic education revealed to be concerned about taking a COVID-19 vaccine (26%) (*p* = 0.002), as well as about its possible effects (62%) (*p* < 0.001). Accordingly, significantly more individuals in this group considered that the vaccine was a mystery to them more frequently (38%) (*p* < 0.001). 

When comparing age groups (Appendix A), there were significant associations found between age group and infection concern (*p* < 0.001): the group of people aged 18 to 24 had fewer concerned participants (9.6%), while the number of individuals aged 55 to 64 had the most (95%), even though this was the group with the smaller percentage of subjects saying they were extremely worried (45%). 

### 3.3. COVID-19 Vaccination Status

There were 152 subjects (about 5% of the total) that were vaccinated at the time of answering the survey. As such, the valid sample for vaccine hesitancy (and related factors) evaluation corresponded to 2938 individuals. 

Of those who were not vaccinated, the majority (91%) would accept the vaccine if available, with the same percentage of acceptance being observed if the vaccine had been recommended by the participant’s employer. 

### 3.4. Cognitive and Emotional Representations of COVID-19 and Vaccine Acceptance

An association was found between vaccine acceptance and perceived worry and knowledge about COVID-19, when adjusting for sex, age, level of education and area of residence (Table 2). 

Participants who were moderately worried about becoming infected were the most likely to accept vaccination (OR = 1.473 [1.048–2.069]; (*p* = 0.026)). Conversely, subjects who were not or were a little worried about infection were the least likely to, given the opportunity, take a vaccine (OR = 0.156 [0.108–0.225]; (*p* < 0.001)).

Respondents who identified themselves as being moderately informed about the COVID-19 pandemic were the most likely to take a vaccine (OR = 1.534 [1.160–2.029]; (*p* = 0.003)). Interestingly, there was no significant difference found between the groups who said they were extremely informed and little or not informed at all (*p* = 0.233).

The influence of COVID-19 representations in vaccine acceptance was the greatest in participants aged 45 to 54, followed by those aged 35 to 44, then 55 to 64, then 25 to 34 and lastly 18 to 24 and >64. There was no difference found between those aged 18 to 24 and those older than 64.

### 3.5. Perceived Vaccine Effectiveness and Vaccine Acceptance 

There was an association between vaccine acceptance and its perceived effectiveness, when adjusting for sex, age, level of education and area of residence (Table 3). 

Counterintuitively, subjects who perceived COVID-19 vaccination to be moderately effective in preventing the disease were significantly more likely to accept it, when compared to the group who deemed it extremely effective (OR = 2.172 [1.787–4.118]; (*p* < 0.001)). Those who thought a vaccine would be little to not effective at all, however, were significantly less likely to accept it (OR = 0.17 [0.009–0.031]; (*p* < 0.001)). Subjects who revealed to be moderately worried about taking a vaccine had a slightly higher risk of not accepting it (OR = 0.550 [0.336–0.900]; (*p* < 0.017)). Those who were not worried were more likely to accept a vaccine (OR = 1.831 [1.178–2.846]; (*p* = 0.007)). Subjects who perceived themselves as not informed about COVID-19 vaccination had a higher likelihood of not accepting it, when compared to those who perceived themselves as extremely informed (OR = 0.508 [0.283–0.913]; (*p* = 0024)). Finally, subjects aged between 35 to 44 were those at greater risk of not accepting a COVID-19 vaccine (OR = 0.356 [0.177–0.716]; (*p* = 0.004)). 

### 3.6. Perceived Vaccine Knowledge and Vaccine Acceptance

When adjusting for sex, age, level of education and area of residence, there were significant differences in vaccine acceptance and perceived knowledge (Table 4).

Individuals who perceived themselves as moderately informed about COVID-19 vaccination were the most likely to accept it (OR = 0.635 [0.455–0.887]; (*p* = 0.008)). Alternatively, there was no significant difference found between the groups who perceived themselves to be extremely informed and little or not informed at all (*p* = 0.736). 

Furthermore, the less an individual considered a COVID-19 vaccine to be a mystery, the more likely they were to accept taking it. Indeed, those who disagreed with the vaccine being a mystery were about 13 times more likely to accept it (OR = 12.875 [8.794–18.848]; (*p* < 0.001)). Participants that neither agreed nor disagreed with this notion were also more accepting of a vaccine (OR = 2.590 [1.799–3.728]; (*p* < 0.001)).

### 3.7. Perceived Vaccine Necessity and Vaccine Acceptance

There was an association found between vaccine acceptance and its perceived importance in the subject’s future health and life, when adjusted for age, sex, education level and area of residence (Table 5).

Compared to the sample group who agreed that their future health would be dependent on a COVID-19 vaccine, the group of people that did not agree with this idea were about 12 times less likely to accept a vaccine (OR = 0.084 [0.050–0.140]; (*p* < 0.001)). Less acceptance was also found within the group that was indifferent to this notion, but to a lesser degree (OR = 0.279 [0.163–0.477]; (*p* < 0.001)). 

Likewise, the odds of not accepting a COVID-19 vaccine were greater in the sample group that disagreed with the idea that their future life would be impossible without a vaccine (OR = 0.390 [0.183–0.831]; (*p* = 0.015)). The influence of vaccine necessity perception on acceptance was greater in participants aged 45 to 54, followed by those aged 35 to 44, then 55 to 64, then 25 to 34 and lastly 18 to 24 and >64. This influence was also more evident in participants with a secondary school level of education (OR = 2.799 [1.085–7.221]; (*p* = 0.033)), when compared to those with a primary school level of education. Having a university level education did not make one more statistically likely to accept vaccination when compared to those with primary level education (*p* = 0.077).

### 3.8. Perceived Vaccine Concern and Vaccine Acceptance 

When analyzing vaccine-related worries and their impact on vaccine acceptance, associations were found between vaccine acceptance and vaccine-related concern, adjusting for sex, age, level of education and area of residence (Table 6). 

Individuals who did not worry about taking a vaccine were 12 times more likely to take it (OR = 12.200 [7.297–20.396]; (*p* < 0.001)). Those who were indifferent to the notion were four times more likely to accept vaccination (OR = 4.001 [2.518–6.356]; (*p* < 0.001)). 

Similarly, those in the group that did not worry about vaccine effects specifically had greater odds of accepting the vaccine, when compared to those who had concerns (OR = 2.717 [1.321–5.588]; (*p* = 0.007)). Subjects indifferent to worrying about vaccine effects were likewise significantly more likely accept vaccination (OR = 2.050 [1.081–3.888]; (*p* = 0.028)).

The influence of vaccine-related worries had a significantly greater influence in male participants’ vaccine acceptance (OR = 0.550 [0.390–0.777]; (*p* < 0.001)).

## 4. Discussion

To the authors’ knowledge, this is the first study that focuses on cognitive and emotional representations of COVID-19 and its vaccines, particularly concerning its impact on vaccine acceptance in a sample of Portuguese participants.

This work’s findings support the already established notion that COVID-19 vaccines are accepted by most people, as found in a Portuguese study conducted three months before the start of the vaccination effort and during its first month [18]. In a European study focusing on vaccine acceptance, the overall willingness to take a COVID-19 vaccine was 73.9% [19]. Likewise, an Australian study found that outright vaccine refusal was only found in 6% of subjects [20]. Concerning Portuguese individuals, in a sample of multiple sclerosis patients, vaccine acceptance was 81%; being receptive to a vaccine was correlated with a subject’s convictions and concerns about COVID-19 and previous vaccination practices [21].

Nevertheless, as highlighted throughout this study, the fact that the prevalence of vaccine refusal is low does not mean that this phenomenon should be neglected. Moreover, as other authors have demonstrated [22] and this study’s findings reinforce, explaining vaccine hesitancy is not straightforward. Indeed, behind this seemingly simple behavior is a very complex interplay of cognitive and emotional dynamics. This is particularly relevant in the COVID-19 context, as evidence suggests that there is a significant group of people that are convinced of the utility of vaccination in general, but hesitant about taking a COVID-19 vaccine [23]. Furthermore, evidence suggests that, as the pandemic has progressed, the number of people intending to refuse vaccination has been increasing [24]. These subjects are of particular interest in what concerns the development of tailored, evidence-based, health communication initiatives. As such, using behavioral models, namely the Self-Regulation Model and the Necessity–Concerns Framework, as done in the development of this work’s questionnaire and interpretation of its results, is paramount.

Given the theoretical framework-guided strategy employed in this study, several pertinent interpretations came about. Firstly, perceiving oneself as extremely informed about the disease does not predict the acceptance of the vaccine. This result may be explained by Dunning–Kruger effect [25]. In fact, subjects with substantial deficits in their knowledge about the vaccine may lack the ability to recognize those flaws and, as such, are predisposed to thinking they understand vaccination when, in reality, they do not. This is particularly relevant in a world where information becomes quickly available, but, at the same time, can be easily manipulated, misinterpreted or simply poorly transmitted.

On the other hand, perceiving COVID-19 vaccines as a mystery is also associated with lower acceptance. This may be, in part, because the notion of “mystery” and “knowledge” are not identical and, as such, one could subjectively think they are very knowledgeable about vaccination, but still associate it with mysterious characteristics. Thus, when evaluating vaccine hesitancy and vaccine knowledge, it may be useful to reframe the question, that is, to attempt to remove the notion of knowledge, which is more vulnerable to biases such as social desirability, and replace it by mystery, which is closely related to worry. Further supporting this hypothesis, this work’s data suggest that concern about taking a COVID-19 vaccine is a determining factor, and men have a greater probability of being affected by it. Regarding sociodemographic factors, being aged between 25 and 65 years old and having a lower education level are both correlated with higher vaccine refusal.

Moreover, it emerged from the findings that people moderately concerned with infection were more likely to accept vaccination than those extremely concerned with it. This seems counter-intuitive at first, and it is not easy to come up with a straightforward explanation. One might posit that individuals with extreme beliefs and concerns are most likely to adopt a perspective distrusting of any outside intervention, especially coming from novel sources, such as COVID-19 vaccination. Avoidance behavior would then act as a defense mechanism against disease that might grow to be maladaptive by making the individual avoid even helpful interventions. The explanation may, however, also lie with concern acting as a positive influence for both anti-disease behavior but conspiracy theory thought processes too. This kind of interplay between thought patterns and behavior is very intriguing and often difficult to confidently describe; future research will surely find it very fruitful to study this dimension.

The present study is not without limitations. First and foremost, its results should be interpreted having in mind the fact that data collection was done entirely online. Thus, as internet access is more limited to people in lower social strata, conclusions regarding this group of people are more limited. Likewise, only people who use the internet and are accessible via social media or online newspaper were potentially informed and answered the survey. These notions are confirmed by verifying that the participant sample is not representative of the whole Portuguese population, as our sample demographics do not correspond to the country as a whole’s demographic characteristics. Another relevant fact to mention is that data collection was done when Portugal was facing the third wave of COVID-19, a new lockdown was imposed, and the number of deaths had reached a new maximum of 1264 per 100,000 inhabitants, on 28 January 2021 [14]. This context may have influenced vaccination perceptions, as this was the subject most discussed in social media and other forums at the time.

## 5. Conclusions

In conclusion, by shedding light on the determinants of COVID-19 vaccine hesitancy, this study provides valuable insights, based on a clear framework, which can be used by policymakers and healthcare professionals in their practice, as well as health educators. Furthermore, it is an important step towards understanding and developing effective, evidence-based health education and science communication strategies in the future, thus contributing to the success of this and other vaccination efforts.

Even though its generalizability is limited, this study’s findings provide future research with fertile ground in the quest to determine how emotional and cognitive pathways influence vaccine hesitancy.

## Figures and Tables

**Table 1 ijerph-19-09268-t001:** Sociodemographic characteristics of the sample (*n* = 3090).

	*n*/Mean (SD)	(%)
Sex		
Female	2205	(71.4%)
Male	882	(28.5%)
Other	3	(0.1%)
Age	42 (15.22)	-
Education level		
Primary school (4th grade)	6	(0.2%)
High school (9th grade)	91	(2.9%)
Secondary school (12th grade)	629	(20.4%)
Bachelor’s degree	1378	(44.6%)
Master’s degree	759	(24.6%)
Doctorate	227	(7.3%)
Employment status		
Unemployed	126	(4.1%)
Housewife	35	(1.1%)
Student	584	(18.9%)
Retired	290	(9.4%)
Employee	1658	(53.7%)
Self-employed	397	(12.8%)
District of residence		
Aveiro	193	(6.2%)
Beja	13	(0.4%)
Braga	189	(6.1%)
Bragança	70	(2.3%)
Castelo Branco	12	(0.4%)
Coimbra	40	(1.3%)
Évora	9	(0.3%)
Faro	25	(0.8%)
Guarda	14	(0.5%)
Leiria	32	(1.0%)
Lisboa	255	(8.3%)
Portalegre	1	(0.0%)
Porto	1877	(60.7%)
Região Autónoma da Madeira	24	(0.8%)
Região Autónoma dos Açores	56	(1.8%)
Santarém	34	(1.1%)
Setúbal	49	(1.6%)
Viana do Castelo	66	(2.1%)
Vila Real	86	(2.8%)
Viseu	45	(1.5%)

**Table 2 ijerph-19-09268-t002:** Cognitive and Emotional Representations of COVID-19 and vaccine acceptance.

	*p*-Value	OR	95% C.I. for OR	
			Lower	Upper
How much has the COVID-19 pandemic affected your life?				
Extremely	Ref			
Moderately	0.808	1.037	0.773	1.393
Little/Not at all	0.897	1.044	0.541	2.016
How worried are you about becoming infected with COVID-19?				
Extremely	Ref			
Moderately	0.026	1.473	1.048	2.069
Little/Not at all	0.000	0.156	0.108	0.225
How informed are you about the COVID-19 pandemic?				
Extremely	Ref			
Moderately	0.003	1.534	1.160	2.029
Little/Not at all	0.233	0.364	0.069	1.919
How much does the COVID-19 pandemic affect you emotionally?				
Extremely	Ref			
Moderately	0.662	1.075	0.777	1.486
Little/Not at all	0.222	0.777	0.517	1.166
Age category				
18–24	Ref			
25–34	0.018	0.513	0.295	0.892
35–44	0.000	0.338	0.204	0.559
45–54	0.000	0.270	0.161	0.452
55–64	0.010	0.461	0.257	0.828
>64	0.490	0.775	0.376	1.598
Sex				
Female	Ref			
Male	0.422	1.137	0.831	1.554
Education level				
Primary school	Ref			
Secondary school	0.304	1.516	0.685	3.354
University	0.280	1.502	0.718	3.140
Country region				
Região Norte				
Grande Porto	0.808	1.054	0.688	1.615
Região Centro	0.947	1.020	0.568	1.833
Grande Lisboa	0.997	0.999	0.575	1.735
Região Sul	0.210	0.544	0.210	1.410
Regiões Autónomas	0.993	1.004	0.387	2.604
Constant	0.000	14.808		

OR—Odds Ratio; Dependent variable: Vaccine acceptance; Independent variables: Perceived life impact of the pandemic, Infection worry, perceived COVID-19 knowledge, COVID-19 pandemic emotional impact, Age category, Sex, Education level, Area of residence; Statistic methods: ENTER: Hosmer–Lemeshow, *p* = 0.760; R^2^ Nalgelkerke *p* = 0.152.

**Table 3 ijerph-19-09268-t003:** Perceived vaccine effectiveness and vaccine acceptance.

	*p*-Value	OR	95% C.I. for OR	
			Lower	Upper
How efficacious do you think vaccination is in preventing COVID-19 infection?				
Extremely	Ref			
Moderately	0.000	2.712	1.787	4.118
Little/Not at all	0.000	0.017	0.009	0.031
How worried are you about taking a COVID-19 vaccine?				
Extremely	Ref			
Moderately	0.017	0.550	0.336	0.900
Little/Not at all	0.007	1.831	1.178	2.846
How informed are you about COVID-19 vaccines?				
Extremely	Ref			
Moderately	0.160	0.753	0.506	1.119
Little/Not at all	0.024	0.508	0.283	0.913
How much would taking a COVID-19 vaccine affect you emotionally?				
Extremely	Ref			
Moderately	0.757	0.921	0.547	1.551
Little/Not at all	0.723	0.924	0.595	1.434
Age category				
18–24	Ref			
25–34	0.175	0.595	0.281	1.259
35–44	0.013	0.423	0.216	0.832
45–54	0.004	0.356	0.177	0.716
55–64	0.060	0.483	0.226	1.032
>64	0.918	0.950	0.358	2.525
Sex				
Female	Ref			
Male	0.312	0.814	0.546	1.213
Constant	0.000	30.827		

OR—Odds ratio; Dependent variable: Vaccine acceptance; Independent variables: Perceived vaccine efficacy, Vaccine worry, Perceived vaccine knowledge, Vaccine emotional effect, Age category, Sex; Statistic methods: ENTER: Hosmer–Lemeshow, *p* = 0.118; R^2^ Nalgelkerke = 0.382.

**Table 4 ijerph-19-09268-t004:** Perceived vaccine knowledge and vaccine acceptance.

	*p*-Value	OR	95% C.I. for OR	
			Lower	Upper
How informed are you about the COVID-19 pandemic?				
Extremely	Ref			
Moderately	0.000	1.730	1.292	2.318
Little/Not at all	0.218	0.377	0.080	1.780
How informed are you about COVID-19 vaccines?				
Extremely	Ref			
Moderately	0.008	0.635	0.455	0.887
Little/Not at all	0.763	0.935	.605	1.446
COVID-19 vaccines are a mystery to me.				
Completely agree/Agree	Ref			
Neither agree nor disagree	0.000	2.590	1.799	3.728
Disagree/Completely disagree	0.000	12.875	8.794	18.848
Age category				
18–24	Ref			
25–34	0.791	1.060	0.687	1.636
35–44	0.668	0.879	0.486	1.587
45–54	0.845	0.944	0.533	1.673
55–64	0.084	0.423	0.159	1.124
>64	0.976	0.985	0.372	2.608
Sex				
Female	Ref			
Male	0.090	0.763	0.558	1.043
Education level				
Primary school	Ref			
Secondary school	0.605	1.231	0.559	2.709
University	0.286	0.673	0.324	1.395
Country region				
Região Norte	Ref			
Grande Porto	0.251	0.719	0.409	1.263
Região Centro	0.055	0.607	0.364	1.011
Grande Lisboa	0.019	0.534	0.316	0.902
Região Sul	0.844	0.942	0.519	1.708
Regiões Autónomas	0.347	1.432	0.677	3.029
Constant	0.004	4.109		

OR—Odds ratio; Dependent variable: Vaccine acceptance; Independent variables: Knowledge (COVID-19, Vaccines, Mystery), Age category, Sex, Education level, Country region; Statistic methods: ENTER: Hosmer–Lemeshow, *p* = 0.142; R^2^ Nalgelkerke *p* = 0.209.

**Table 5 ijerph-19-09268-t005:** Perceived vaccine necessity and vaccine acceptance.

	*p*-Value	OR	95% C.I. for OR	
			Lower	Upper
My future health will depend on COVID-19 vaccination.				
Completely agree/Agree	Ref			
Neither agree nor disagree	0.000	0.279	0.163	0.477
Disagree/Completely disagree	0.000	0.084	0.050	0.140
My life will be impossible without a COVID-19 vaccine.				
Completely agree/Agree	Ref			
Neither agree nor disagree	0.235	0.602	0.260	1.393
Disagree/Completely disagree	0.015	0.390	0.183	0.831
Without a COVID-19 vaccine, I will become very sick.				
Completely agree/Agree	Ref			
Neither agree nor disagree	0.943	0.959	0.308	2.992
Disagree/Completely disagree	0.213	0.496	0.164	1.496
Age category				
18–24	Ref			
25–34	0.026	0.511	0.283	0.922
35–44	0.001	0.401	0.232	0.695
45–54	0.000	0.281	0.160	0.491
55–64	0.007	0.417	0.222	0.783
>64	0.615	0.802	0.339	1.898
Sex				
Female	Ref			
Male	0.644	0.923	0.658	1.295
Level of education				
Primary school	Ref	Ref		
Secondary school	0.033	2.799	1.085	7.221
University	0.077	2.206	0.917	5.307
Country region				
Região Norte	Ref	Ref		
Grande Porto	0.416	1.212	0.763	1.925
Região Centro	0.715	1.127	0.594	2.137
Grande Lisboa	0.261	1.415	0.772	2.594
Região Sul	0.116	0.431	0.151	1.232
Regiões Autónomas	0.212	2.094	0.656	6.679
Constant	0.000	120.251		

OR—Odds ratio; Dependent variable: Vaccine acceptance; Independent variables: Vaccine need (Health-dependency, Life possibility, Future sickness), Age category, Sex, Education level, Country region; Statistical methods: ENTER: Hosmer–Lemeshow, *p* = 0.949; R^2^ Nalgelkerke *p* = 0.279.

**Table 6 ijerph-19-09268-t006:** Perceived vaccine concern and vaccine acceptance.

	*p*-Value	OR	95% C.I. for OR	
			Lower	Upper
Taking a COVID-19 vaccine worries me.				
Completely agree/Agree	Ref			
Neither agree nor disagree	0.000	4.001	2.518	6.356
Disagree/Completely disagree	0.000	12.200	7.297	20.396
I’m worried about COVID-19 vaccination’s effects.				
Completely agree/Agree	Ref			
Neither agree nor disagree	0.028	2.050	1.081	3.888
Disagree/Completely disagree	0.007	2.717	1.321	5.588
COVID-19 vaccines are a mystery to me.				
Completely agree/Agree	Ref			
Neither agree nor disagree	0.076	1.426	0.964	2.109
Disagree/Completely disagree	0.000	2.753	1.855	4.085
Age category				
18–24	Ref			
25–34	0.312	0.730	0.396	1.343
35–44	0.575	0.853	0.490	1.486
45–54	0.126	0.644	0.367	1.132
55–64	0.910	1.038	0.547	1.966
>64	0.126	1.847	0.842	4.053
Sex				
Female	Ref			
Male	0.001	0.550	0.390	0.777
Education level				
Primary school	Ref			
Secondary school	0.264	1.582	0.708	3.535
University	0.890	0.949	0.453	1.989
Country region				
Região Norte	Ref			
Grande Porto	0.901	0.971	0.611	1.543
Região Centro	0.860	0.944	0.500	1.784
Grande Lisboa	0.913	0.967	0.532	1.760
Região Sul	0.186	0.494	0.174	1.405
Regiões Autónomas	0.553	1.369	0.485	3.870
Constant	0.238	1.809		

OR—Odds ratio; Dependent variable: Vaccine acceptance; Independent variables: Vaccine concern (Vaccine worry, Vaccine effects, Mystery) Age category, Sex, Education level, Country region; Statistical methods: ENTER: Hosmer–Lemeshow, *p* = 0.986; Nalgelkerke R^2^ = 0.279).

## Data Availability

All participants were informed about the study objectives and data collection procedures.

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
