# Peer review of "Acceptance and Adherence to COVID-19 Vaccination—The Role of Cognitive and Emotional Representations"

_ijerph, 2022, doi:10.3390/ijerph19159268_

Round 1

Reviewer 1 Report

Please see the "response to authors" file below for the authors review 

Author Response

Please find our response in the annexed file.

Respectfully, 

The authors.

Reviewer 2 Report

This is an important and interesting paper on the role of cognitive and emotional representations related to vaccination and their influence in vaccination hesitancy of COVID'19 in Portugal.

Abstract: needs more focus when presenting the findings. For example regarding women authors indicate that they reported they pandemic affected their lives, were more concerned with infection and reported higher emotional disturbance. What about gender differences in vaccine hesitancy, given the gender disparities presented above?

Authors report that people in the age category 25-65 were more likely not to accept vaccination. This is a very broad age category, in particular that the risk of contagion and serious ill is more pronounced for people over 60.

Literature review, the authors relied in their study on the necessity concerns framework. Can you focus on it and elaborate more? 

I think it will be of great help to the reader to present the framework and to write the specific hypothesis of the study. In that way the results will be more organized (according to clear hypothesis) and will be easier for the reader to follow.

Regarding the inclusion criteria of the study, is this parallel to the age category that was called to take the vaccine in the same day?

In the findings section, when discussing the findings you need to indicate in which table you presented them so the author  can follow you presentation of the findings when he/she looks at the corresponding table.

again if you present hypothesis the reading of the paper will be easier.

Author Response

(The authors gave the same response as above.)

Round 2

Reviewer 1 Report

The authors are to be congratulated on the excellent and appropriate revisions that address all this reviewers issues in the first iteration. The article will be a nice addition to the Global Health aspect of the journal and opens interesting insights into personal health decisions that should be evaluated in other populations 

On page 12 however the last highlighted sentence in section 4 needs revision. "... our sample demographics are do not correspond...." revise by eliminating the "are" then it makes sense

Author Response

We have corrected the manuscript as noted, and thank the reviewer for noticing what was a writing lapse. 

Finally, we would like to deeply thank the reviewer for their valuable contribution in imporving our work. Hopefully they are as happy with the final result as we are, and we will continue our research efforts certain that we are on a good path.

Reviewer 2 Report

the author/s have revised the manuscript. I do not have further comments

Author Response

We are happy to hear that the reviewer is satisfied with our corrections. We also extend our deep gratefulness to their work on helping us improve our manuscript.